# Occurrence and Distribution of Tetrabromobisphenol A and Diversity of Microbial Community Structure in the Sediments of Mangrove

**DOI:** 10.3390/biology12050757

**Published:** 2023-05-22

**Authors:** Yongcan Jiang, Qiang Wang, Yunling Du, Dong Yang, Jianming Xu, Chongling Yan

**Affiliations:** 1PowerChina Huadong Engineering Corporation Ltd., Hangzhou 311122, China; 2College of Environmental and Resource Sciences, Zhejiang University, Hangzhou 310058, China; 3College of the Environment and Ecology, Xiamen University, Xiamen 361102, China; 4State Key Laboratory of Herbage Improvement and Grassland Agro-Ecosystems, College of Pastoral Agriculture Science and Technology, Lanzhou University, Lanzhou 730000, China

**Keywords:** tetrabromobisphenol A, mangrove sediments, occurrence and distribution, microbial diversity, correlation analysis

## Abstract

**Simple Summary:**

Tetrabromobisphenol A has been used extensively in fire-resistant materials in recent decades, and scientists have extensively investigated its toxic effects on organisms and humans. The present study aims to investigate the distribution characteristic of tetrabromobisphenol A in three mangrove swamps; analyze the associated microbial community structure involved in the tetrabromobisphenol A degradation; and explore the relationships between the microbial community and environmental factors. The results indicated mangrove sediments from the Zhangjiang Estuary, Jiulongjiang Estuary, and Quanzhou Bay in Southeast China contained high levels of tetrabromobisphenol A. Correlation analysis revealed that total organic carbon plays a critical impact on tetrabromobisphenol A distribution in the mangrove sediments. The microbial community structures of the three mangrove sediments were similar, while the taxonomic profile of their sensitive responders differed markedly. The genus *Anaerolinea* was dominant in the mangrove sediment and may be responsible for the in situ dissipation of tetrabromobisphenol A. Redundancy analysis indicates that combining tetrabromobisphenol A, total nitrogen, and total organic carbon would induce a shift in the microbial community structure of mangrove sediments. This work provided insight into the endogenous microbial community diversity of different mangrove sediments which may be one of the factors contributing to the different tetrabromobisphenol A concentrations.

**Abstract:**

The occurrence and distribution characteristics of tetrabromobisphenol A (TBBPA) and its relationship with microbial community diversity in different mangrove sediments need further investigation. The results of this study indicated levels of TBBPA in mangrove sediments from the Zhangjiang Estuary (ZJ), Jiulongjiang Estuary (JLJ), and Quanzhou Bay (QZ) in Southeast China ranging from 1.80 to 20.46, 3.47 to 40.77, and 2.37 to 19.83 ng/g dry weight (dw), respectively. Mangrove sediments from JLJ contained higher levels of TBBPA, possibly due to agricultural pollution. A correlation analysis revealed a significant correlation between total organic carbon (TOC), total nitrogen (TN), and TBBPA distribution in ZJ and JLJ mangrove sediments, but not in QZ mangrove sediments. TOC significantly affected the distribution of TBBPA in mangrove sediments, but pH had no effect. High-throughput 16S rRNA gene sequencing showed that *Pseudomonadota* dominated the sediment bacteria followed by *Chloroflexota*, *Actinobacteota*, *Bacillota*, *Acidobacteriota*, *Bacteroidota*, and *Aminicenantes* in mangrove sediments. Although the microbial community structure of the ZJ, JLJ, and QZ mangrove sediments was similar, the taxonomic profile of their sensitive responders differed markedly. The genus *Anaerolinea* was dominant in the mangrove sediments and was responsible for the in situ dissipation of TBBPA. Based on redundancy analysis, there was a correlation between TBBPA, TOC, TN, C/N, pH, and microbial community structure at the genus level. Combining TBBPA, TN, and TOC may induce variations in the microbial community of mangrove sediments.

## 1. Introduction

Tetrabromobisphenol A, the most prevalent of the brominated flame retardants, is used in electronic devices, textile processing, polyurethane foams, and other industries to prevent the spread of fire [1,2]. There have been various studies on the occurrence of TBBPA in natural environments, such as water, biota samples, soil, and sediments [3]. In addition to sites associated with the manufacturing and processing of TBBPA-based materials, the main TBBPA emission sources are e-waste recycling sites and wastewater treatment plants [4], with the highest reported TBBPA concentration registering up to 732 mg/kg dw in wastewater sewage sludge [5]. Previous studies have shown that TBBPA can disrupt thyroid and estrogen hormone function, as well as induce liver toxicity, cytotoxicity, and neurotoxicity [6]. Due to its toxicity, high hydrophobicity, and lipid solubility, TBBPA can easily be accumulated and transported in the food chain via different mediums, which could potentially pose a threat to human health [7]. However, the current lack of better alternatives means that TBBPA will continue to be used for a long time. 

Mangrove wetlands, located along tropical, subtropical, and warm temperate coastlines, are a significant blue carbon sink, playing an important role in the biogeochemical cycles of nitrogen and phosphorus in coastal environments [8]. Owing to the rapid urbanization and industrialization of coastal regions, mangrove ecosystems often receive large amounts of pollutants as a result of human activities [9]. It has been reported that mangrove sediments act as an important sink for a variety of halogenated flame retardants, primarily because of their organic carbon content, anaerobic properties, and hypersaline conditions [10]. Furthermore, high concentrations of polycyclic aromatic hydrocarbons (PAHs) have been widely detected in mangrove wetland sediments from around the world [11]. Other studies have shown that mangroves play a crucial role in the retention of microplastics from both marine and terrestrial inputs [12]. However, the TBBPA concentration in mangrove sediments has only been reported for areas of the Pearl River Estuary (China) and Singapore [13,14]. Therefore, an analysis of surface mangrove sediments collected from different locations is needed to explore the abundance and spatial characteristics of TBBPA therein. 

Researchers have studied TBBPA removal from environmental media using advanced oxidation techniques, physical adsorption, physicochemical degradation, and biodegradation [15,16]. Bioremediation is recognized as an eco-friendly technique to remove pollutants from the environment. TBBPA degradation via microbially mediated reductive debromination has been demonstrated in anaerobic soil, sediment, and sewage sludge [17,18]. The methylation of TBBPA in aerobic sediments yields methyl ether derivatives that are more lipophilic than the parent, so they accumulate more in biological systems [19]. The study by Xu et al. reported that the strain *Rhodococcus jostii.* RHA1 can transform and mineralize TBBPA to less toxic products under aerobic conditions [20]. An isolated marine strain of *Pseudomonas aeruginosa* was demonstrated to degrade TBBPA in cometabolic degradation processes [21]. A novel isolated strain, *Enterobacter* sp. T2, that effectively degrades TBBPA was shown to contain key genes for debromination and the later mineralization steps [22]. Several studies have demonstrated that the amendment of TBBPA-degrading bacteria into soil or sediment stimulates the removal efficiency of TBBPA in microcosms [23]. However, research is limited on the diversity of indigenous microbial communities that degrade TBBPA in situ. 

Mangrove sediments are subject to long-term contamination by heavy metals and organic pollutants from terrestrial and marine sources, and changes in the microbial biomass and community structure may occur in sediments after being exposed to environmental pollutants [24,25]. Additionally, mangrove sediments contain many nutrients that support the survival of a diverse microbial community. The composition and abundance of the sediment microbial community have a significant impact on the degradation efficiency of organic pollutants. Previous studies have reported a shift in the microbial community structure using phospholipid fatty acid analysis after exposure to polybrominated diphenyl ethers (PBDEs) and PAHs [26,27]. Wang et al. found that *Sphingomonas*- and *Mycobacterium*-like populations were significantly associated with PAHs, while the *Dehalococcoides*-like population was significantly related to the PBDE congener in mangrove sediment microcosms [28]. Another study indicates that microplastics affect the diversity and richness of microorganisms in mangrove sediments [29]. These studies, which were carried out in the laboratory, mainly focused on the response of the microbial community structure to organic contaminants, but research in the real environment is still infrequent. Thus, field studies are expected to better explore the links between organic pollutants and the microbial community structure.

The occurrence and distribution characteristics of TBBPA and the structure of the associated microbial community in mangrove sediments are still unknown. Furthermore, the above-described studies have left knowledge gaps with respect to the relationship between microbial composition and environmental characteristics. Hence, the objectives of the present study were to investigate the occurrence and distribution characteristics of TBBPA in different mangrove wetlands; analyze the associated microbial community structure involved in TBBPA degradation; and explore the relationships between the microbial community, TBBPA, and environmental factors using redundancy analysis. The findings presented herein will broaden our knowledge of the spatial characteristics and microbial mechanisms of intrinsic TBBPA removal in mangrove sediments. 

## 2. Materials and Methods

### 2.1. Study Area and Sediment Collection

Surface sediments were collected between October and December 2018 from the ZJ estuary mangrove wetland (21 samples), JLJ estuary mangrove wetland (26 samples), and QZ Bay mangrove wetland (23 samples) (Figure 1). The predominant mangrove species in these wetlands are *Aegiceras corniculatum*, *Avicennia marina*, and *Kandelia obovata*. The ZJ, JLJ, and QZ mangroves have areas of 2360 hm^2^, 420 hm^2^, and 400 hm^2^, respectively. The ZJ estuary mangrove forests are the most well-preserved and suffer low-to-moderate sewage discharge, agricultural pollution, and urbanization [30]. In the QZ Bay mangroves, there is moderate-to-heavy human pressure, including the exploitation of fisheries, rapid urbanization, and textile pollution [31]. JLJ mangrove is threatened by moderate-to-severe anthropogenic exploitation such as increasing aquaculture, agricultural pollution, and tourism. The fresh mangrove sediment samples (0–20 cm) were collected from the three locations using precleaned stainless-steel spoons; then, they were freeze-dried and placed in a small refrigerator to be brought back to the laboratory and stored at −20 °C until experimental analyses were performed. 

### 2.2. Pretreatment of Samples

A portion of the fresh sediment samples was dried at room temperature and subsequently sieved through 0.85 mm nylon mesh to remove stones and coarse debris. After collecting the sieved samples, one part was stored in polyethylene bags to measure the pH; the other part was passed through a nylon sieve of 0.149 mm and then also stored in polyethylene bags to measure the contents of TOC and TN and perform an extraction analysis of TBBPA. The other portion of the fresh sediment was homogeneously mixed and then passed through a 2.0 mm sieve to remove large sand grains and plant debris before microbial community analysis [32].

### 2.3. Physical and Chemical Analysis

#### 2.3.1. Analysis of Selected Sediment

The pH of the selected sediments was measured at a sediment/water ratio of 1:2.5 using a pH meter (PHS-2F, Shanghai, China). The TOC and TN contents were analyzed with an Element Analyzer Vario EL (Elementar Analysensystem GmbH, Hanau, Germany) in CHNS mode. The total organic carbon (TOC) content was determined with the same equipment after the removal of carbonate carbon using 10% HCl. 

#### 2.3.2. Extraction and Purification of TBBPA

Precisely 2.5 g of dried sediment was spiked with 50 ng of ^13^C_12_-TBBPA (50 mg/mL in methanol, 99%, CIL, USA) as a surrogate standard. The spiked sample was kept at room temperature in darkness for 24 h to allow the carrier solvent methanol to completely evaporate before the extraction and purification of TBBPA. An accelerated solvent extractor (Dionex ASE 350, Thermo Fisher Scientific, Waltham, MA, USA) was used to extract the sample with dichloromethane (DCM)/acetone (2/1, *v*/*v*, TEDIA, Fairfield, OH, USA). The extraction conditions are described in detail in the Supporting Information. Each extract was cleaned by eluting through an ENVI-Carb SPE cartridge (Supelco, Sigma-Aldrich, St. Louis, MO, USA) using a modified method based on previous work [33]. Following cleanup, the eluent was concentrated to dryness under a gentle stream of nitrogen gas, dissolved in 500 mL of methanol (Merck, Darmstadt, Germany), and filtered using a 0.22 µm PTFE syringe filter before ultra-performance liquid chromatography–tandem mass spectrometry (UPLC-MS/MS) analysis. 

#### 2.3.3. Instrumental Analysis of TBBPA

To quantify the TBBPA content via the select ion mode, UPLC-MS/MS (1290 Infinity UPLC, 6490 Triple Quadrupole Mass Spectrometer, Agilent, Santa Clara, CA, USA) was coupled with the electrospray ionization negative mode. For liquid chromatography separation, a ZORBAX Eclipse Plus C_18_ column (Rapid Resolution High Definition, 2.1 × 100 mm, 1.8 µm, Agilent) was used with a front guard column (ZORBAX Eclipse Plus C_18_, 2.1 × 5 mm, 1.8 mm, Agilent). The mobile phase was composed of Milli-Q water (mobile phase A) and acetonitrile: methanol (30:70, *v*/*v*, mobile phase B; Merck, Germany). The gradient steps are described in Appendix A. The mobile phase flowed at 0.3 mL/min, and the sample injection volume was 1.0 mL. The mass spectrometry operation parameters are outlined in Appendix A. The chromatograms of TBBPA detected with UPLC-MS/MS are presented in Appendix A.

### 2.4. DNA Extraction and PCR Amplification

The fresh sediments collected from the ZJ, JLJ, and QZ mangroves were subjected to total DNA extraction using the E.Z.N.A.^®^ soil DNA Kit (Omega Bio-tek, Norcross, GA, USA) following the manufacturer’s protocol. The concentrations of the extracted and purified DNA from each sample were measured using a NanoDrop 2000 UV-Vis spectrophotometer (Thermo Scientific, Wilmington, NC, USA), and DNA quality was checked with 1% agarose gel electrophoresis. The V3-V4 hypervariable region of the bacterial 16S rRNA gene was amplified using the universal primers 338F and 806R on a thermocycler PCR system (GeneAmp 9700, ABI, USA) [34]. The primers used for the amplification of gene sequences are provided in Appendix A, and details of the PCR amplification procedure are provided in the Appendix A. Purified amplicons from each sample were normalized to equimolar concentrations and paired-end sequenced (2 × 300 bp) on the Illumina MiSeq platform (Illumina, San Diego, CA, USA) from Majorbio Bio-Pharm Technology Co., Ltd. (Shanghai, China) according to the standard protocol. QIIME2 was used to analyze the Illumina MiSeq-generated data. The taxonomy of each 16S rRNA gene sequence was assigned using the RDP Classifier algorithm against the SILVA (SSU138) 16S rRNA database at a confidence threshold of 70%.

### 2.5. Quality Assurance and Statistical Analysis

Experimental set reagent blank, spiked blank, matrix spiked, and parallel were used to ensure the reliability of extraction and purification procedures for TBBPA analysis. No targets were detected in the procedural blank sample. The TBBPA recovery rates of the blank addition and matrix addition in the sediment were 76.18 ± 6.91% and 78.71 ± 6.71%, respectively. The surrogate standard (^13^C_12_-labeled TBBPA) was spiked into each sample before the extraction and purification steps. Recovery of the surrogate standard ^13^C_12_-TBBPA from the sediment was 87.04 ± 12.19%. The final TBBPA concentrations in the samples were corrected using the recoveries. The background concentrations of TBBPA in the ZJ, JLJ, and QZ mangrove sediments were 0.15, 0.21, and 0.18 ng/g, respectively. The method detection limit of TBBPA based on a signal-to-noise ratio of 3:1 was 0.08 ng/g for sediment. 

OriginPro 2016 (Origin Lab, USA) and SPSS Statistics 20 (IBM SPSS, USA) were used to analyze the data. One-way analysis of variance (ANOVA) was performed to test statistically significant (p < 0.05) differences among treatments. A distance-based redundancy analysis (db-RDA) was used to investigate the effects of soil physicochemical indicators on soil bacterial community structure. All microbial data were analyzed in the auspicious cloud platform (https://cloud.majorbio.com. accessed on 5 October 2021.).

## 3. Results

### 3.1. Occurrence and Distribution of TBBPA in Mangrove Sediments

The concentrations of TBBPA in each mangrove sediment sample are summarized in Appendix A. The TBBPA detection rate was 100% in the three mangrove sediments. The TBBPA concentrations in sediments from the ZJ, JLJ, and QZ mangroves ranged from 1.80 to 20.46, 3.47 to 40.77, and 2.37 to 19.83 ng/g dw, with respective average and median concentrations of 7.78 and 8.64 ng/g dw for ZJ, 11.63 and 8.43 ng/g dw for JLJ, and 9.67 and 8.80 ng/g dw for QZ, as shown in Table 1 and Appendix A. These results show that the mangrove sediments at the three locations were contaminated by TBBPA. It is noted that the concentrations of TBBPA at sites S1, S2, and S3 in the sediments of the JLJ mangrove were the highest, but the median concentrations here were the lowest, indicating that these concentrations varied greatly, which may be related to the environment of the sampling sites (Appendix A, Appendix A). The average level of TBBPA contamination in JLJ mangrove sediments was higher than that in ZJ Estuary and QZ Bay mangrove sediments (Appendix A). 

### 3.2. Correlation between TBBPA and Environmental Factors

The correlation between TBBPA (a hydrophobic organic contaminant) and TOC, TN, and pH in soil/sediment was investigated. Appendix A summarize the pH and TOC and TN contents in each mangrove sediment sample. The range and mean values of TOC in the mangrove sediments were, respectively, 1.18–2.94% and 1.98% for ZJ, 1.02–2.39% and 1.59% for JLJ, and 1.01–2.16% and 1.30% for QZ (Table 1). In general, the TOC contents followed the order: ZJ > JLJ > QZ. The Pearson correlation coefficient analysis of TOC and TBBPA in the three sediments showed that the TBBPA concentration of the ZJ Estuary sediments had a significant correlation with the TOC (r^2^ = 0.47, *p* = 0.003). Similarly, the TBBPA concentration in the JLJ Estuary sediments also exhibited a good correlation with the TOC (r^2^ = 0.35, *p* = 0.002). In contrast, no significant correlation was found between TBBPA and TOC in the QZ Bay Estuary sediments (r^2^ = −0.04, *p* = 0.76) (Figure 2). The range and mean values of TN in the mangrove sediments were, respectively, 0.10–0.25% and 0.15% for ZJ, 0.10–0.18% and 0.13% for JLJ, and 0.09–0.15% and 0.11% for QZ (Table 1). Correlation analysis showed that TN in the mangrove sediments of ZJ, JLJ, and QZ correlated well with TOC (Figure 2). The correlation coefficient value r was 0.84, 0.55, and 0.76 in the ZJ, JLJ, and QZ mangrove sediments, respectively (*p* < 0.05). The results indicate that the TOC and TN in the sediments might have the same source and distribution pattern. There were no significant correlations between the pH and TBBPA in the mangrove sediments, and the average pH of the mangrove sediments at ZJ, JLJ, and QZ were 6.64 ± 0.08, 6.68 ± 0.15, and 6.64 ± 0.12, respectively. 

### 3.3. Microbial Community Diversity

High-throughput sequencing was performed using the DNA from nine mangrove sediment samples collected from the ZJ, JLJ, and QZ mangroves. In total, 412,668 high-quality sequences were generated, with 45,852 sequences per sample. The sequences were clustered at a 97% similarity level into a total of 9436 operational taxonomic units (OTUs), which were classified into 61 phyla, 158 classes, 303 orders, 544 families, 1052 genera, and 2298 species (Appendix A). The trends in the Shannon-index-based rarefaction curves increased with increasing reads and plateaued when the number of reads reached 45,000, and Good’s coverage values were all above 96% (Appendix A, Appendix A). Thus, this demonstrates that the data are representative of the microbial diversity in each sediment sample. The microbial α-diversity indices analysis shows that there was no significant correlation between the Chao, ACE, and Sorb index in ZJ, JLJ, and QZ sediments (Figure 3A). OTU-based hierarchical clustering and UniFrac-based principal component analysis (PCA) showed that the microbial community structures in the mangrove sediments were not significantly different among the three wetlands (Figure 3B and Appendix A) (*p* > 0.05). The first two principal components (PCs) explained 34.72% of the total OTU variance (*p* = 0.18), with PC1 explaining 20.21% of the variance and separating the OTUs of the JLJ and QZ mangrove wetlands. PC2 explained 14.51% of the variance and separated samples from the ZJ and QZ mangrove wetlands. 

### 3.4. Microbial Community Composition in Mangrove Sediments

Microorganisms play a key role in the removal of pollutants in environmental sediments, but most cannot be cultured in the laboratory. The dominant microorganisms in three mangrove sediments were analyzed, and the main phyla were *Pseudomonadota*, *Chloroflexota*, *Actinobacteota*, *Bacillota*, *Acidobacteriota*, *Bacteroidota*, *Aminicenantes*, *Nitrospirota*, *Gemmatimonadota*, *unclassified-k-norank*, *Gracilibacteria*, *Nitrospinota*, *Cyanobacteria*, *Planctomycetota*, and *Spirochaetota* (Figure 4A). The total relative abundance of these main phyla in ZJ, JLJ, and QZ mangrove sediments reached 91.63%, 92.56%, and 91.83%, respectively (Appendix A). The major microbial *genera* (relative abundance > 1%) in the sediments were *Anaerolinea*, *Sulfurovum*, *Chloroflexus*, *Acidobacterium*, *Thiobacillus*, *Nitrospira*, *Desulfobulbus*, *Clostridium*, *Gracilibacterium*, *Actinobacterium*, *Thioalkalispira*, *Robiginitalea*, *SJA-68*, *Belgica2005-10-ZG-3*, and *OM1_clade* (Figure 4B). The total relative abundance of the main genus was 51.83%, 49.72%, and 50.39% in the ZJ, JLJ, and QZ mangrove sediments, respectively (Appendix A). These results indicate that there are abundant microorganisms in mangrove sediments, which may contribute to the in situ degradation and mineralization of halogenated organic compounds. 

### 3.5. Dynamics of Microbial Community

Microbial community members associated with the degradation of aromatic and halogenated organic substances are present in different mangrove sediments [9]. The relative abundance of *Pseudomonadota* and *Chloroflexota* in the JLJ, ZJ, and QZ mangrove sediments was 41.92% and 17.88%, 37.55% and 21.54%, and 36.31% and 21.03%, respectively (Appendix A). The composition of the top 30 genera (relative abundance > 0.01) was analyzed for the sediments (Figure 5). The genus abundance of *Anaerolinea* was significantly higher than the other genera (*p* < 0.05): 10.33%, 10.08%, and 8.78% in the ZJ, QZ, and JLJ mangrove sediments, respectively (Appendix A). In the ZJ sediment, the relative abundances of *Anaerolinea*, *unclassified-p Chloroflexi*, *Gracilibacteria*, *Robiginitalea*, *Thioalkalispira*, and *Xanthomonadales_Incertae_Sedis* were higher than in the other two sediments (Figure 5). Additionally, the relative abundance of *Sulfurovum* was significantly higher in the ZJ sediment than the other sediments (Appendix A). In the QZ sediment, the relative abundance of *Acidobacteria* was higher than the JLJ and ZJ sediments; in the JLJ sediment, the relative abundance of *Nitrospira* and *Desulfobulbus* was higher than in the other two sediments (Figure 5). Although the sediments from the ZJ, JLJ, and QZ wetlands had a similar microbial community structure, the taxonomic profile of the sensitive responders differed from one another (Figure 5 and Appendix A). 

### 3.6. Correlations of Microbial Community with Environmental Characteristics

Mangrove sediments can be impregnated with TBBPA, which may explain the differences between the microbial communities in the three mangrove wetlands. Thus, the correlations between microbial communities, environmental factors, and TBBPA in mangrove sediments were examined using redundancy analysis (Figure 6). TBBPA, TOC, TN, C/N, and pH in the sediments explained 58.98% of the variability in the microbial community structure of all canonical axes (RDAs). The first canonical axis (RDA1) explained up to 39.71% of the variance at the genus level (r^2^ = 0.20, *p* = 0.05). According to our results, this is a correlation between TBBPA, TOC, TN, C/N, pH, and the structure of microbial communities at the genus level in mangrove sediments. The variability in the genus-level composition was significantly correlated with TN (r^2^ = 0.36, *p* = 0.025), TOC (r^2^ = 0.35, *p* = 0.026), and TBBPA (r^2^ = 0.22, *p* = 0.049). Thus, TN, TOC, and TBBPA were the characteristics which were the most impactful on the microbial community structure in the sediment. The present study is the first to report correlations of the microbial community with TBBPA, TOC, TN, C/N, and pH in the in situ mangrove environment. 

## 4. Discussion

The concentrations of TBBPA in the ZJ, JLJ, and QZ mangrove sediments were higher than those previously detected in sediments of the Mandai mangrove (Singapore) and Futian mangrove (China) [35,36], while the TBBPA concentration of the JLJ mangrove sediment was similar to that reported for the Tantou mangrove (China) [13] (Table 2). Some former studies suggest that mangroves in the JLJ Estuary are more significantly affected by agricultural pollution and that the JLJ Estuary mangrove wetland is widely contaminated with PAHs, polychlorinated biphenyls (PCBs), and PBDEs [37]. Zhang et al. studied the level and distribution of halogenated organic pollutants in JLJ mangrove sediments, reporting that the concentrations of dichlorodiphenyltrichloroethane (DDT) and PBDEs were 21.00–84.00 and 9.00–66.00 ng/g dw, respectively [11]. Furthermore, their analysis showed that the halogenated organic pollutants mainly originated from agricultural pollution emissions [11]. Thus, the level of TBBPA contamination, one of the most widely used brominated flame retardants, in the JLJ mangrove sediment was significantly affected by agricultural pollution emission. 

Previous studies have shown that the levels of brominated flame retardants and TBBPA contamination are related to the amount of urbanization and industrialization, as well as economic and anthropogenic activities in the area [17,33,38,39,40,41,42]. Some historical data showing TBBPA concentrations in sediment, soil, and sewage sludge around the world are shown in Table 2. The TBBPA concentration in the soil at an e-waste disposal plant in Vietnam ranged from not detectable to 2900.00 ng/g dw [43]. TBBPA was detected in the soil of the Qingyuan town e-waste disposal plant (the largest in China) and its surrounding farmland, at concentrations ranging from 84.00 to 646.04 ng/g dw [43]. In Shouguang City, the main TBBPA production location in China, TBBPA was detected in the surrounding soil at concentrations ranging from 1.64 to 7758.00 ng/g dw, with an average concentration as high as 672.00 ng/g dw [45]. In the Pearl River Estuary, which is one of the most economically developed areas in China, TBBPA contamination is commonly detected in tributary sediments. Specifically, TBBPA in the surface sediments of the river sediment reportedly ranges from 0.10–304.00 ng/g dw, respectively [46]. In addition, the TBBPA concentration in the surface sediment of the Dongjiang River increased from nd–82.30 in 2006 to 3.80–230.00 ng/g dw in 2009 [47]. The ZJ Estuary mangrove forests suffer low-to-moderate sewage discharge and urbanization [31]. In the QZ Bay mangroves, there is moderate-to-heavy human pressure, including the exploitation of fisheries as well as rapid urbanization [30]. In these two mangrove areas, industrialization and urbanization are the main factors that affect TBBPA pollution levels in the sediments. 

It has been revealed that environmental factors play a very important role in the sorption, partitioning, and bioavailability of hydrophobic organic contaminants in both sediment and soil [48]. Studies have shown that sediment organic carbon plays a very important role in the adsorption, distribution, and bioavailability of hydrophobic organic matter [49,50]. Some studies have reported a good correlation (r^2^ = 0.70–0.96) between organohalogens (PBDEs and DDT) and TOC at 0.10–5.37% TOC in river sediments [51]. Luo et al. demonstrated that PBDE and TBBPA concentrations in surface sediments were significantly correlated with the TOC content, but PBDE and TBBPA contamination in sediments were also related to pollutant emissions from sources within the area [52]. Alternatively, the TBBPA concentration in fishing port sediments was strongly associated with the local population density but weakly correlated with the sediment TOC content [53]. This indicates that TOC has an important impact on the distribution behavior of TBBPA in mangrove sediments. Guerra et al. indicated that alkaline sediments facilitate the release of TBBPA from particles and that TBBPA is easily adsorbed on particle surfaces under acidic conditions [40]. In general, the contents of TBBPA, TOC, and TN in the sediment were significantly positively correlated, while the sediment pH had no influence on the TBBPA concentration distribution. 

Microorganisms play a key role in the removal of pollutants in environmental sediments, but most cannot be cultured in the laboratory [54,55]. The diversity and structure of the indigenous microbial community are important for the debromination of halogenated organic pollutants in marine and mangrove sediments [56,57]. In microcosms containing mangrove sediments, halogen-reducing bacteria were found, and the bacteria with the potential to reduce were *Gammaproteobacteria*, *Actinobacteria*, *Clostridia*, *Flavobacteria*, and *Bacilli*, where reductive dehalogenation was observed in sediments through BDE-209 [58]. *Sphingomonas*, *Mycobacterium*, and *Dehalococcoides*, all belonging to the phyla *Pseudomonadota* and *Actinobactota*, have previously been identified as the main bacteria performing PAH and PBDE contamination removal in mangrove sediments [58,59]. *Planctomycetota* and *Gemmatimonadota* distribution was also detected in the mangrove sediments. There was also a presence of *Planctomycetota* and *Gemmatimonadota* in the sediments; these organisms have also been identified as being actively involved in the utilization of organic carbon [60]. In previous studies, an analysis of bacteria in mangrove sediments contaminated with petroleum hydrocarbons and PAHs revealed that the main genera were *Alcaligenes*, *Bacillus*, *Flavobacterium*, *Micrococcus*, and *Pseudomonas aeruginosa* [61,62]. According to Pan et al.’s study on *Dehalococcoides* spp., dehalogenating bacteria in mangrove wetlands reduce and dehalogenate BDE-47 in sediments [26]. It has been found that the important mechanism for microbial reductive dehalogenation removes halogenated organic pollutants from mangrove sediments [63,64]. In these sediments, both TBBPA and PBDEs are halogenated organics; thus, potential TBBPA-degrading phyla are present. 

Studies have shown that *Anaerolinea* is able to anaerobically biodegrade TBBPA [65]. Both *Sulfurovum* and *Thiobacteria* have been detected in mangrove sediments stimulating anaerobic TBBPA degradation [62]. *Chloroflexi* strains have frequently been identified as the key dehalogenators in sediments contaminated with chlorinated organics [56]. Researchers have previously examined the debromination ability of OHRB, including *Geobacter* spp., *Dehalobacter* spp., *Dehalococcoides* spp., *Dehalococcoides* spp., and *Dehalomonas* spp. [66]. The abundances of *Acidobacteria*, *unclassified-k-norank*, *Aminicenantes, SJA-68*, *Actinobacteria*, *Gaiellales*, *SBR2076*, *Desulfobacca*, and *Cyanobacteria* in the QZ mangrove sediment were higher than in the JLJ and ZJ mangrove sediment. It has been reported by Peng et al. that *cyanobacteria* can utilize TBBPA as a carbon source for growth and may also be useful for the bioremediation of mangrove sediment contaminated with TBBPA [67]. Relative to the ZJ and QZ mangroves, the JLJ sediment exhibited higher abundances of *Thiobacillus, Nitrospira, Desulfobulbus, Clostridium_sensu_stricto_1, Xanthomonadales*, *Gammaproteobacteria*, *Gemmatimonadetes*, and *Gemmatimonadaceae* (Appendix A). Activated sludge containing nitrifiers also nitro-debromated TBBPA [68]. The reductive debromination of TBBPA might act as a terminal electron acceptor for *Desulfobulbus* and *Desulfobacca*, which are sulfur-reducing bacteria [57]. Above all, the differing endogenous microbial community diversity may be one of the factors contributing to differences in the TBBPA concentrations of the different mangrove sediments. 

Wu et al. indicated that the combined pollution of heavy metals, PAHs, and PBDEs would best illustrate the variation in the microbial community of river sediments from Guiyu town, China [10]. Xiao et al. suggested that heavy metals, such as Cr, Pb, and V, together with organic matter fractions might participate and shape the fungal communities in mangrove sediments [69]. Significant relationships between the concentrations of halogenated flame retardants and the microbial community structure in such sediments, as well as TBBPA, syn-DP, and BTBPE, appeared to have the most significant impact on the microbial community structure [13]. Moreover, the results showed that BDE-154 was the key pollutant acting on the microbial community structure in mangrove sediments [27]. Data from another study revealed that soil microorganism growth and composition were associated with TBBPA exposure time and concentration [70]. Redundancy analysis showed that organic carbon, nitrate, and TN significantly influenced the bacterial community structure in the soil of the Loess Plateau [71]. The results of our study confirm that TBBPA, TOC, and TN may work together to change the composition of microbial communities in mangrove sediments. Future research efforts should aim to understand the mechanisms of interactions between microbial communities and their environment, as well as identify novel microorganisms and their metabolic pathways. 

## 5. Conclusions

Sediments from three mangroves (ZJ, JLJ, and QZ) pervasively contaminated with TBBPA were analyzed; the TBBPA levels ranged from 1.80–20.46, 3.47–40.77, and 2.37–19.83 ng/g dw, respectively. The contamination in sediment from the JLJ was significantly influenced by agricultural pollution discharges. The results showed that the TBBPA concentrations in the ZJ and JLJ sediments significantly correlated with TOC. TOC has an important impact on the distribution behavior of TBBPA in mangrove sediments. These studied sediments are rich in microorganisms, some of which might be associated with the degradation and mineralization of TBBPA. The main genus *Anaerolineae* contributed considerably to the dissipation of TBBPA in all three mangrove sediments. Redundancy analysis highlighted the significant relationship between TBBPA, TOC, TN, and the microbial community structure at the genus level. Our results suggest, for the first time, that the combined presence of TBBPA, TOC, and TN may have the potential to alter the structure of microbial communities in mangrove sediments.

## Figures and Tables

**Figure 1 biology-12-00757-f001:**
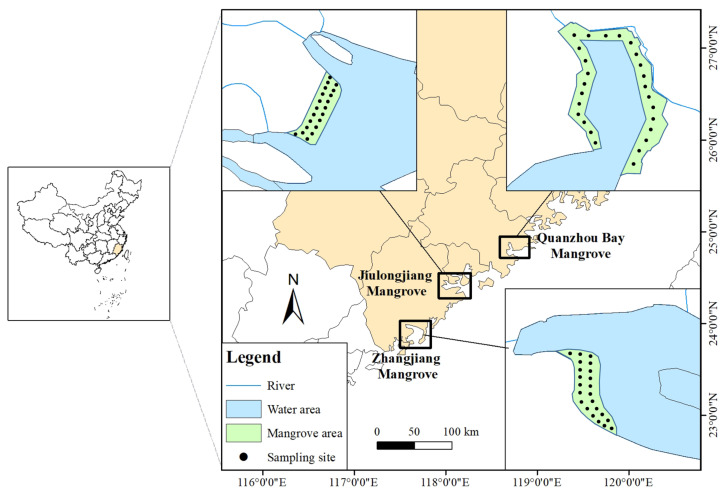
Map of sampling sites of the ZJ, JLJ, and QZ sediment. ZJ, JLJ, and QZ indicate Zhangjiang Estuary mangrove, Jiulongjiang Estuary mangrove, and Quanzhou Bay mangrove, respectively.

**Figure 2 biology-12-00757-f002:**
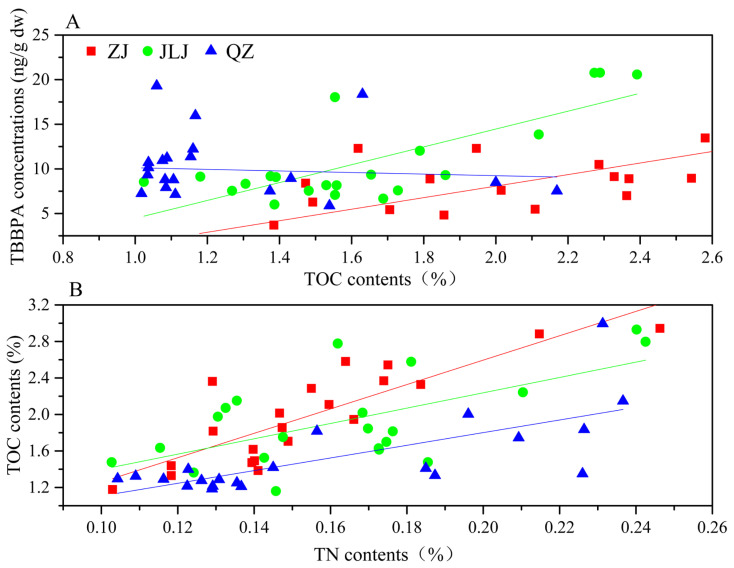
Relationships between TOC and TBBPA (**A**), and TN contents (**B**) in ZJ, JLJ, and QZ sediments. ZJ, JLJ, and QZ indicate Zhangjiang Estuary mangrove, Jiulongjiang Estuary mangrove, and Quanzhou Bay mangrove, respectively.

**Figure 3 biology-12-00757-f003:**
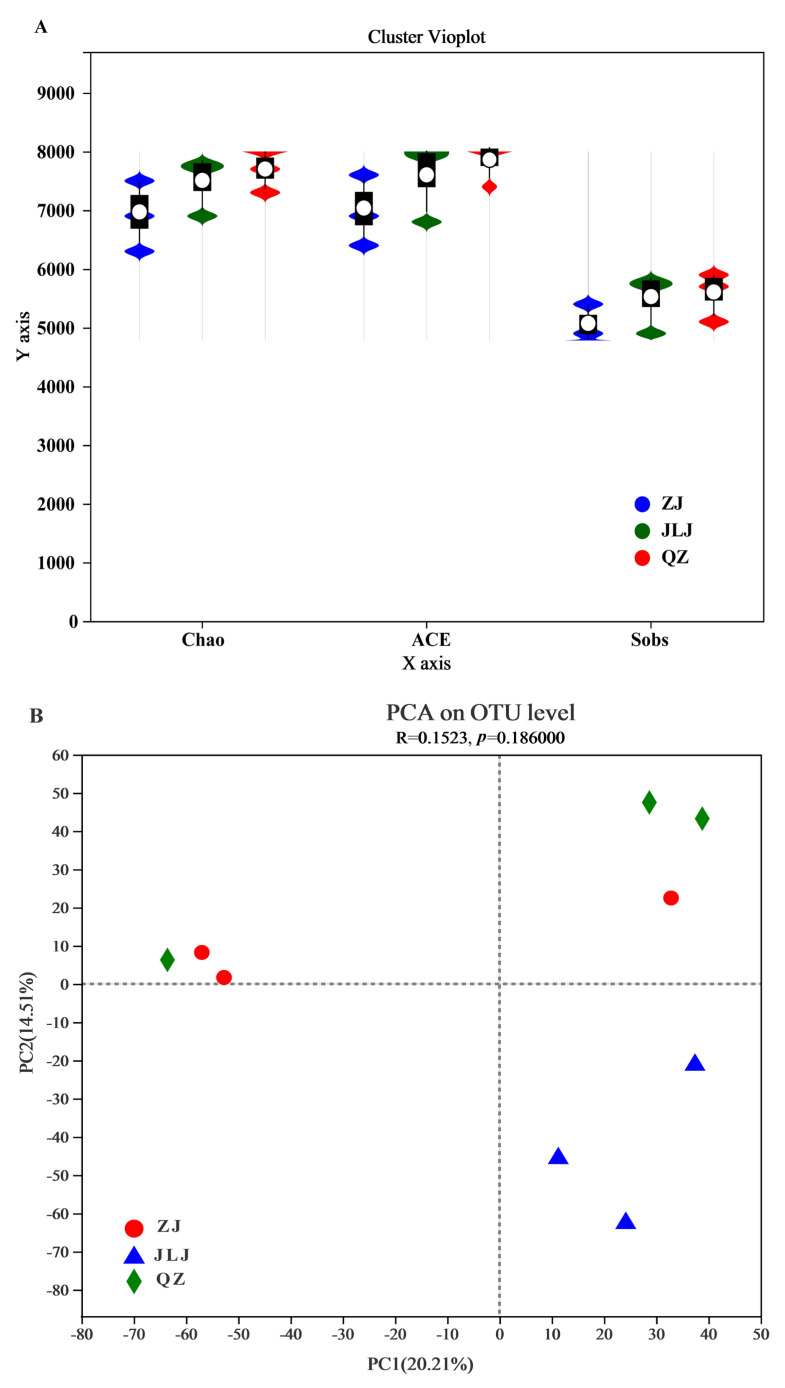
(**A**) Microbial α-diversity indices analysis between the Chao, ACE, and Sorb indices in ZJ, JLJ, and QZ sediments. (**B**) OUT level-based principal component analysis (PCA) showing the difference of bacterial community between ZJ sediments, JLJ sediments, and QZ sediments. ZJ, JLJ, and QZ indicate Zhangjiang Estuary mangrove, Jiulongjiang Estuary mangrove, and Quanzhou Bay mangrove, respectively.

**Figure 4 biology-12-00757-f004:**
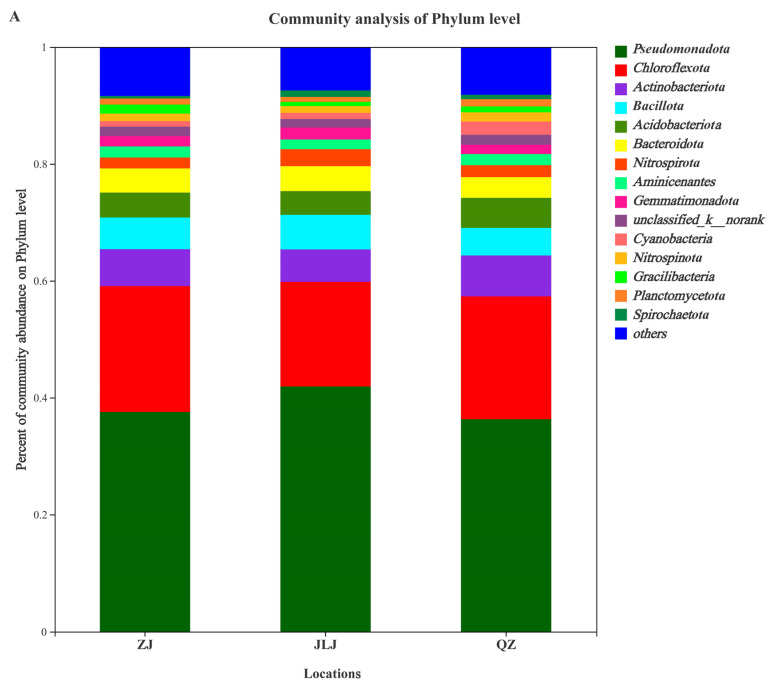
(**A**) Phylum level of microbial community composition and (**B**) Genus level of microbial community composition in the sediments of ZJ, JLJ, and QZ. (relative abundance > 0.01). ZJ, JLJ, and QZ indicate Zhangjiang Estuary mangrove, Jiulongjiang Estuary mangrove, and Quanzhou Bay mangrove, respectively.

**Figure 5 biology-12-00757-f005:**
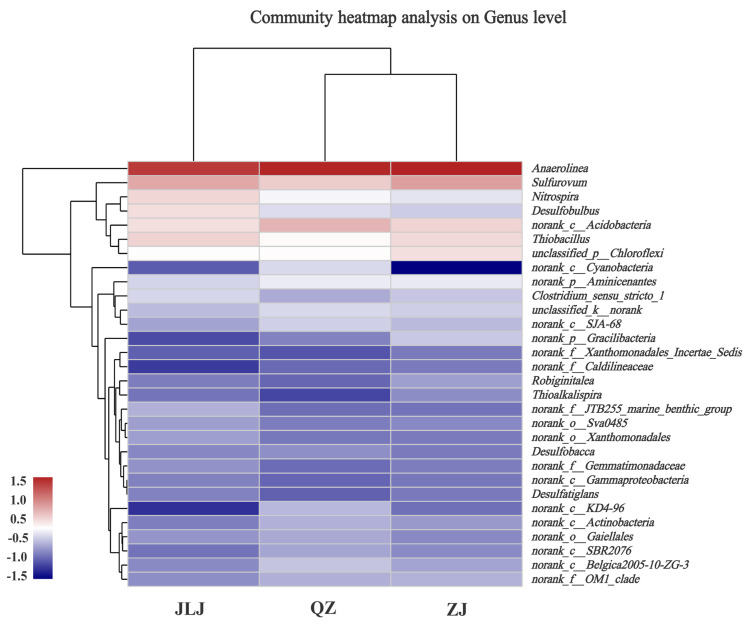
Hierarchical cluster analysis showing the relative abundances of top 30 genera (relative abundance > 0.01) in sediments of ZJ, JLJ, and QZ mangroves. The color-coded panel indicates the changes of relative abundance. ZJ, JLJ, and QZ indicate Zhangjiang Estuary mangrove, Jiulongjiang Estuary mangrove, and Quanzhou Bay mangrove, respectively.

**Figure 6 biology-12-00757-f006:**
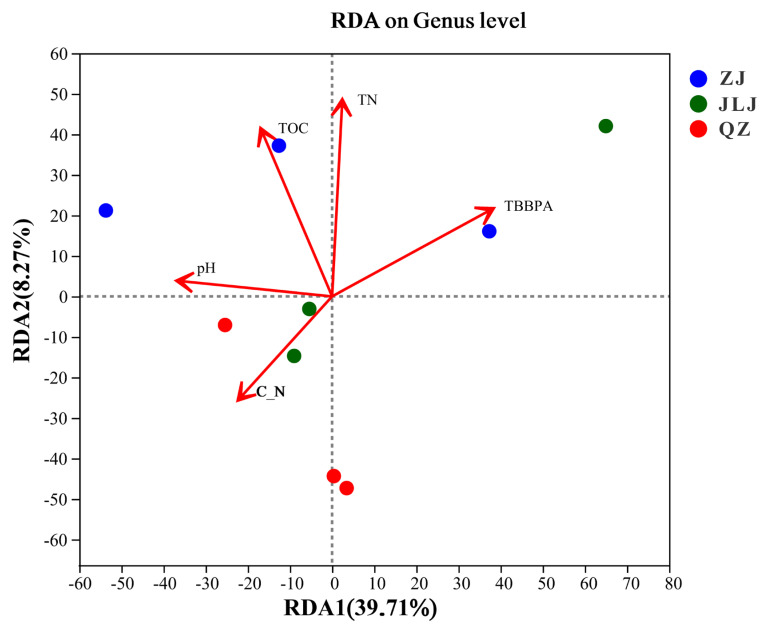
Redundancy analysis between microbial community and environmental characteristics and TBBPA in ZJ, JLJ, and QZ sediments. ZJ, JLJ, and QZ indicate Zhangjiang Estuary mangrove, Jiulongjiang Estuary mangrove, and Quanzhou Bay mangrove, respectively.

**Table 1 biology-12-00757-t001:** Concentrations of TBBPA in mangrove sediments from ZJ Estuary, JLJ Estuary, and QZ Bay Estuary, Southeast China (ng/g dw).

	Zhangjiang Estuary (*n* = 21)			Jiulongjiang Estuary (*n* = 26)			Quanzhou Bay Estuary (*n* = 23)		
	Range	Median	Mean	Range	Median	Mean	Range	Median	Mean
TBBPA	1.8–20.46	8.43	7.78	3.47–40.77	8.80	11.63	2.37–19.83	8.64	9.67
TOC ^a^	1.18–2.94%	1.90	1.98%	1.02–2.39%	1.54	1.59%	1.01–2.16%	1.14	1.30%
TN ^b^	0.10–0.25%	0.14	0.15%	0.10–0.18%	0.13	0.13%	0.09–0.15%	0.10	0.11%
pH ^c^	6.49–6.84	6.71	6.64	6.38–6.89	6.67	6.68	6.43–6.85	6.66	6.64

Notes: ^a^: Total organic carbon (%); ^b^: total nitrogen (%); ^c^: pH in sediment; dw: dry weight.

**Table 2 biology-12-00757-t002:** Levels of TBBPA in soil/sediment/sewage sludge from different locations around the world (ng/g dw).

Locations	Sample Types	TBBPA Levels	References
Mandai mangrove, Singapore	Surface sediment	0.048–0.22	[34]
Futian mangrove, Shenzhen	Surface sediment	0.30–1.85	[35]
Tantou mangrove, Guangzhou	Surface sediment	0.16–37.50	[13]
Osaka, Japan	River sediment	5.00–140.00	[37]
Israel, Ramat-Hovav	Desert soil	nd–450 × 10^6^	[17]
Netherlands	Sewage sludges	2.00–600.00	[38]
Madrid, Spain	Industrial soil	3.40–32.20	[33]
Northeast of Spain	Sewage sludges	nd–1329.00	[39]
Catalonia, Spain	Sewage sludges	nd–472.00	[40]
Ningbo, the east of China	Surface soil	0.025–78.60	[41]
Bui Dau, Vietnam	Surface soil	5.00–2900.00	[42]
Qinyuan, Guangdong	Surface soil	84.00–646.04	[43]
Shouguang, Shandong	Surface soil	1.64–7758.00	[44]
Dongjiang River, Guangzhou	surface sediment	3.80–230.00	[14]
Pearl River Delta, China	Surface sediment	0.06–304.00	[45]
Dongjiang River, Guangzhou	Sediment cores	7.90–450.00	[46]

*Notes*: nd: not detected; dw: dry weight.

## Data Availability

Data are available on request.

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
