# Peer review of "Occurrence and Distribution of Tetrabromobisphenol A and Diversity of Microbial Community Structure in the Sediments of Mangrove"

_biology, 2023, doi:10.3390/biology12050757_

Round 1

Reviewer 1 Report

This manuscript investigated the levels of TBBPA and the microbial community in 3 sampling locations. Overall, the manuscript is well-constructed and easy to read. However, this study is a survey and quite descriptive. Hypothesis should be strongly indicated, and the rationale supporting them should be provided accordingly. Since the detection of particular microbial taxa in an impacted environment is quite common, including analysis of the TBBPA functional genes would extend our knowledge on this issue.

Please see additional comments below.

 1. Some toxic compounds at a low level can be commonly detected in impacted mangrove sediments as a background concentration. The authors claimed that mangrove sediments from the three locations were contaminated by TBBPA, so the background concentration of TBBPA from the non-impacted sites should be provided. Also, is the level of TBBPA at ng/g dw considered contamination? The significant difference in the TBBPA levels across 3 locations should be tested for. Then, the obtained information should be used to explain the difference in microbial patterns in the following section.

2. For each sampling site, the samples were corrected from several locations (Fig. 1), but the results of the microbial community (sections 3.3 and 3.4) showed only one sample per sampling site? Please further clarify. Replication should be considered here? To minimize bias from DNA extraction and sequencing, replicate samples should be conducted. Moreover, all raw sequencing data should be deposited in the Genbank database. Genbank accession number should be provided accordingly.

3. According to section 3.5, the key message involves the dynamics of the microbial community in 3 sediment samples. Why is “Functional Capacity” relevant here?

4. Why was RDA, not CCA, used to study correlations between microbial community and environmental factors? Are there any assumptions that support the methodology used? Moreover, including more samples from each location for microbial analysis would strengthen the findings. Other than TBBPA, why only TOC, TN, and pH were analyzed in this study? Are there any other environmental factors that should be included to help explain the findings?

5. Since this study is a survey of TBBPA and the microbial community, it would be interesting if the authors add the significance/benefit of the study in the abstract.

Line 120: Were the samples preserved during transportation?

Lines 221-227: Since TBBPA is considered a subset of TOC, why was the correlation between them negative?

Lines 231-232: Can the authors really mention that the TOC and TN were from the same sources?

Lines 251-254: When the authors mentioned that the microbial community structures in the mangrove sediments were significantly different among the three wetlands, it should be supported by statistical analysis.

Lines 258-259: Can the authors really mention this?

Line 272: What does “unclassified-k-norank” mean?

Lines 28-282: Why the resulting 16S rRNA gene sequencing could indicate in situ degradation and mineralization of TBBPA? These findings should be supported by an analysis of TBBPA functional genes?

Lines 290-291: Any evidence to support this statement? Microbial taxa capable of degradation of toxic compounds are generally detected in impacted environments.

Line 318: Why was the resulting RDA considered “a strong correlation”?

Figure 1 looks good, but it is confusing when zooming into each location. The orientation of the extended part should be exactly the same as the main figure.

Figures 2, 4, and 5 have a low resolution. Please revise.

Moderate editing of English language would help improve the quality of the manuscript.

Reviewer 2 Report

1. In general, the introduction and justification for the study could be improved to more than "the necessity of further investigation". The rationale for looking specifically TBBPA pollution in the particular selected environment is not completely clear.

2. Specific comments:

Line 275: Authors mentioned main genera and then proceed to mention some taxa not corresponding to that level of taxonomic classification. 

Line 280: The presented results do not necessarily indicate the presence of microorganisms involved in TBBPA degradation as the authors assure. Further analysis (e.g. metabolic inference)  could be performed based on the taxonomic diversity data. Consider removing text from 280-282 in this paragraph and leave this statement only for the discussion. 

Line 442: Missing a word in the phrase "interactions between microbial and their environment".

Line 456-458:  To conclude about the modification of the microbial communities related to TBBPA pollution and TOC and TN, goes out of the capacities of the study and experimental design since there´s no way to have a non-contaminated control sample. Definitely, there might be an effect of these factors and you can hypothesize about the alterations in the bacterial composition, but to conclude it in that manner is not correct. 

Overall, good use of the language and the text is easy to understand. There are some minor errors to correct. 

Reviewer 3 Report

Authors present manuscript about the microbial community structure in the mangrove sediments. While introduction and discussion parts are written concisely, the exposition of the microbial analysis itself is very crude and needs some adjustment.

Here are some particular comments:

Line 21, 28, please use tenses consistently (past, not present)

Line 23 upgrade taxonomy here and in the text according to valid phyla names (Proteobacteria -> Pseudomonadota, etc.) doi:10.1099/ijsem.0.005056

Line 46 “consequently posing a potential threat to human beings” – please specify this quite vague phrase.

Line 47 though phrase “to be used and abused for a long time to come” is really great, I don’t think that it suits academic language.

All in all very thorough introduction.

Line 131 at which temperature were stored the samples between collecting and microbial community analysis?

Line 138 what do you mean by “the same equipment”?

Line 167 how many samples from each location were used for dna extraction?

Line 173 add citation for these primers

Line 181 – current Silva version is 138. Why did you use 123 version?

Line 197 “All microbial data analysis in the auspicious cloud 197 platform “ – sentence is missing a verb.

Sequencing data processing description in the methods section is quite undetailed. It should be referencing supporting information more clearly.

Line 212 are differences in TBBPA content between sites statistically significant?

Line 221 please specify confidence threshold (p-value) for the pearson coefficient analysis. Why did you use both r and R2?

Line 241 how did you chose 3 samples for each site for sequencing analysis? Mark them in the tables S4-S6

Line 250 correlation between what did you want to show by alpha-indices?

Line 252/263 Figure 3B caption and manuscript text states that you used PCoA. But the graph heading states “PCA”, which is a different analysis. Which did you actually use?

Line 254 Figure 3B suggests that your samples were mixed up between location (green and red dots).

Line 275 in the list of “microbial genera” you mention a whole variety of taxonomical ranks (genus, phylum, family, etc).

Line 279 what do you mean by the “main genus”?

Line 280 how exactly the previous sentence indicates that “there might be abundant microorganisms related to TBBPA degradation in mangrove sediment”?

Figure 4A and figure 5 are mutually redundant. Pick one.

It is better toy use color-blind palettes for all your figures.

Line 296 how did you measure significance of differences in abundance of OTUs between sites?

Line 214 The sentence meaning is not clear.

Line 70-77 of the supporting information talks about PICRUSt2 analysis. But the text of the manuscript contains no results of the functionality of the microbiomes of the mangrove sediments.

Diskussion lacks some clarity at which bacteria found in the mangrove sediments are connected to TBBPA degradation. 

Sometimes strange choses of phrases were made.

Tenses schould be used more consistently.

Round 2

Reviewer 1 Report

Overall, the quality and clarity of the manuscript have improved.